# Volunteer Food Handlers' Safety Knowledge and Practices in Implementing National School Nutrition Programme in Gauteng North District, South Africa

**Paul K. Chelule * and Mavis Ranwedzi**

Department of Public Health, School of Healthcare Sciences, Sefako Makgatho Health Sciences University, Molotlegi Street, Ga-Rankuwa 0208, South Africa
* Correspondence: paul.chelule@smu.ac.za; Tel.: +27-12-521-3330

**Abstract:** Volunteer food handlers (VFHs) working in school feeding programmes contribute immensely to the safety of food served to the school learners. However, their level of knowledge and practice of safe food handling is questionable. This study investigated food safety knowledge and practices of VFHs working for the National School Nutrition Programme (NSNP) in Gauteng North District, South Africa. This was a descriptive quantitative study in which data were obtained using a standard structured questionnaire. A total of 115 VFHs participated in the study. Most of the VFHs (n = 84, 73.0%) had secondary education with working experience of between 1 to 12 months. Although a high level of knowledge on food safety was demonstrated by the VFHs, this did not fully translate into safe food handling practices. The new VFH recruits were 66% more likely to wear protective clothing than the rest (OR: 0.34, 95%CI: 0.12–0.91, *p* = 0.033). Women were 90% more likely to wear aprons than men (OR = 0.11; 95 CI: 0.03–0.45; *p* = 0.002). In this study, VFHs demonstrated adequate levels of knowledge on food safety. Lack of facilities contributed to VFH non-compliance to safe food handling practices. Thus, there is a need to further support VFHs to comply with recommended safe food handling practices.

**Keywords:** volunteer food handlers; school feeding programme; food safety; food hygiene; school children; practices; knowledge; training

## 1. Introduction

Annually, it is reported that many South Africans contract food poisoning, and children are the most vulnerable [1]. For example, the South African department of health registered 2560 cases of foodborne diseases, where the majority of casualties (numbering 1700) were learners in primary and secondary schools [1,2]. The prevalence of food poisoning is greater in children, who are more at risk as their body systems are not fully developed [3]. The most frequent foodborne disease is Salmonella food poisoning, which presents as acute gastroenteritis in the affected patients [4,5]. Insufficient knowledge on proper food handling may contribute to food poisoning outbreaks such as this [2]. The identified risk factors of foodborne illnesses include poor cooking processes, unsuitable storage temperatures, cross-contamination, unhygienic food handling and obtaining food from unclean sources, as outlined by the WHO guidelines [6]. Thus, food prepared to feed children should be of high-quality standards and free of contamination.

In South Africa, school-going children from economically disadvantaged communities often attend school hungry, as they have little or nothing to eat at home. One of the strategies initiated by the government to address the problem was to introduce a National School Nutrition Programme (NSNP) in 1994 [7]. This programme provides food to children from economically disadvantaged communities during the day at school. This is considered the most important meal of the day, as most children have little or nothing to eat before and after school [8]. The benefits of implementing the programme include pupil increased

enrolment, school attendance, punctuality, retention and academic performance [9]. In South Africa, the NSNP reaches over 9 million learners in 21,400 primary and secondary schools nationally, under the Department of Basic Education administration [8]. The program is implemented by volunteer food handlers (VFHs), who prepare and serve food to the learners. The VFHs are the sole custodians of food preparation as there are no other employed permanent staff to undertake catering services in the schools. The DBE's approved ratio of VFH is 1 food handler to every 200 learners [10].

The suitability of the food preparation facilities and VFHs to serve food in schools is not well known, as there are scanty reports on the awareness of food safety and food handling practices by VFHs in South Africa. A number of studies have reported on unsafe food handling practices, but in different areas and communities [11,12]. This could have led to reported food poisoning outbreaks in public in schools, including those covered by the NSNP [2,13]. One research report conducted in South Africa and Ghana showed that infrastructure design and kitchen facilities in the schools with feeding schemes were not designed for meal preparation [14]. Such improper infrastructure can lead to food contamination. However, this report is older than 10 years. Thus, a thorough investigation is required to establish the contributing factors to these problems in order to formulate effective interventions. This study aimed to assess the awareness on food safety and food handling practices of VFHs in schools implementing the National School Nutrition Programme at the Gauteng North District in South Africa.

## 2. Materials and Methods

This research employed a quantitative cross-sectional study design. The study was conducted in schools undertaking the NSNP in Gauteng North District. Gauteng North District has a total of 38 schools (27 primary schools and 11 secondary schools) participating in NSNP and the number of learners in the 2015/16 financial year was 28,849. The DBE's determination of the number of food handlers' appointment ratio is 1: 200 (1 food handler to approximately 200 learners), with a total of 144 food handlers serving a total of 28,849 learners in all the schools. Food handlers working in all the schools were targeted for the interview since this population was small. All participants of the study voluntarily consented to participate, and this was substantiated by their signing of the consent forms.

The researcher collected data using a standard structured questionnaire designed in English, as all VFHs were English literate. The tool was designed purposefully for this study using information from the literature, which employed validated tools in similar studies [2,3,11,12] and consisted of five sections, namely background information of the school, demographic characteristics of food handlers, awareness of food handlers on safe food handling, food handling practices and an observation checklist to determine the availability, condition and suitability of the cooking facilities.

Data were collected using a researcher-administered questionnaire, which largely comprised close-ended questions. Data collection was conducted during school hours, with the consent of the school principal, as it was the only period that the food handlers were available. On the day of the visit, VFHs were requested to participate upon signing their informed consent forms. The data were analysed using STATA 13 statistical software (StataCorp LLC, College Station, TX, USA). Descriptive statistics were used to calculate, interpret and report the data. Bivariate logistic regression was used to determine the relationship between socio-demographics, level of knowledge and safe food handling practices. Statistical significance was deduced and *p*-values less than 0.05 were considered significant.

The ethical clearance to conduct the study was obtained from Sefako Makgatho University Research Ethics Committee (SMUREC/H/287/2016:PG). The ethical clearance certificate was issued before the commencement of the study. In addition, permission to conduct the study in schools was sought and granted from the Gauteng Department of Education. The researcher requested institutional permission from the school heads. Informed consent was sought from the VFHs, who were also allowed to withdraw their participation in the study at any time they wished.

## 3. Results and Discussion

### 3.1. Socio-Demographic Characteristics of Volunteer Food Handlers

This study investigated the knowledge and food safety practices of VHFs in schools conducting a school feeding program in Gauteng Province, South Africa. A total of 115 out of 144 volunteer food handlers from 38 schools in the study setting participated in the study, resulting in a response rate of 80%. Their ages ranged between 41 and 50 years. Most of the VFHs (73.0%) had secondary education. Almost all the respondents (91.3%) were married females with work experience between 1–12 months (81.7%). More than half (60.9%) of the VFHs had also been trained through attendance in at least one of the workshop categories (training period ranging from one day to two weeks) organised by the DBE (Table 1). Furthermore, over 90% of the VFHs acquired their food handling skills through observation of the more experienced food handlers.

**Table 1.** Socio-demographic characteristics of volunteer food handlers.

| Characteristic | N | % |
|:---:|:---:|:---:|
| **Gender** | | |
| Female | 105 | 91.3 |
| Male | 10 | 8.7 |
| **Age category** | | |
| 20–30 | 15 | 13.1 |
| 31–40 | 32 | 27.8 |
| 41–50 | 45 | 39.1 |
| >51 | 23 | 20.0 |
| **Marital status** | | |
| Single | 8 | 7.0 |
| Married | 58 | 50.4 |
| Widowed | 5 | 4.4 |
| Divorced | 44 | 38.3 |
| **Education** | | |
| Never went to school | 4 | 3.5 |
| Primary school | 27 | 23.5 |
| Secondary school | 84 | 73.0 |
| **Work experience** | | |
| 1–12 months | 94 | 81.7 |
| >12 months | 21 | 18.3 |
| **Workshop attendance** | | |
| 1–3 days | 70 | 60.9 |
| 4–7 days | 4 | 3.5 |
| 1–2 weeks | 1 | 0.9 |
| Other (never attended workshop) | 40 | 34.8 |

In this study, over 90% of food VFHs were female. This is similar to other South African studies citing 70–100% female VFHs in their setting [10,11]. This shows that females seem to be more amenable to volunteering than their male counterparts to undertake this task. Given the history of South Africans, most men work far from their families, and this makes women remain at home to take care of the children. Therefore, opportunities like these will mostly be available to women who reside in the vicinity of the schools. This could also be related to cultural issues, where African societal norms ascribe food preparation to women at home rather than men [11,12,15]. The study results also showed that over 80% of VFHs had a less than one year appointment period at the school. This is due to the NSNP guidelines that require VFHs to rotate at the end of each financial year in order to allow equal opportunities to other members in the community. The gap on the knowledge and food safety practices could be attributed to the rotational model that the department perpetuates. However, the fact that most VFHs had secondary school level education also could be an advantage, as this makes it easy to train them on safe food handling practices.

### 3.2. Volunteer Food Handlers' Knowledge on Food Safety

Volunteer food handlers were asked a number of questions to assess their knowledge on food safety. The majority of VFHs (97.4%) acknowledged that safe food handling practice was an important part of their job responsibilities. In addition, on their awareness on health and safety, most (94%) acknowledged that VFHs should not prepare food if they were sick with flu or diarrhoea, as doing so would potentially make the students sick (80.9%). When asked the importance of training in improving food safety practices, 98.3% agreed that training was important and necessary. More than 90% of the VHFs acquired knowledge on food safety practices through observation, and the rest obtained it through training. For those who were trained, most of them (60%) had attended a 1–3 day training during their employment period. Additionally, 97.4% of the volunteer food handlers believed that formal training on food safety was necessary for food safety purposes, while the rest responded otherwise. Furthermore, on responses to questions on food labelling and checking expiry dates on food products, there was an overwhelming response (99.6%) that it was their responsibility to check. Thus, the respondents correctly knew that it was important to check expiry dates of ingredients before using them.

On the assessment of knowledge on their personal hygiene, the respondents demonstrated high level of knowledge (Table 2). For example, almost all of them (99%) intimated that it was necessary to wash hands after visiting the toilet, and that one could not wear jewellery when preparing food (86%). Thus, the study findings demonstrate VFHs' adequate knowledge regarding proper food handling and food safety.

**Table 2.** Volunteer food handlers' knowledge on food safety.

| Response Questions | Yes (%) | No (%) |
|---|---|---|
| **Knowledge on training** | | |
| Acquired knowledge though observation | 108 (93.9) | 7 (6.1) |
| Acquired knowledge through training | 7 (6.1) | 108 (93.9) |
| Food safety training is important and necessary | 112 (97.4) | 3 (2.6) |
| Attending workshops leads to improved food safety | 113 (98.3) | 2 (1.7) |
| Certification of food practice is necessary | 113 (98.3) | 2 (1.7) |
| It is the responsibility of food handlers to make sure that foods served to learners are safe | 109 (90.8) | 6 (5.2) |
| Safe food handling is an important part of my job responsibilities | 112 (97.4) | 3 (2.6) |
| Pathogens can be found in the skin of a healthy-looking food handler | 56 (48.7) | 59 (51.3) |
| It is necessary to put on kitchen gloves when cooking | 39 (33.9) | 76 (66.1) |
| **Knowledge on food labelling and storage** | | |
| It is important to check food expiry dates | 114 (99.1) | 1 (0.9) |
| Food with latest expiry dates should be consumed first | 86 (74.8) | 29 (25.2) |
| Is it safe to store cleaning products in the kitchen | 84 (73.0) | 31 (27) |
| Keeping food in warm place increases the risk of contamination | 94 (81.7) | 21 (18.2) |
| **Knowledge on health and safety practices** | | |
| Food handlers should regularly be medically examined | 110 (95.6) | 5 (4.4) |
| You can cook when you are sick with flu | 21 (18.3) | 94 (81.7) |
| You can cook when you are sick with diarrhoea | 21 (18.3) | 94 (1.7) |
| Preparing food when you are sick can make learners sick | 93 (80.9) | 22 (19.1) |
| Food handlers with unhygienic practices could be the source for food contamination with food poisoning pathogens | 104 (90.4) | 11 (9.6) |
| Food handlers can be the source of food poisoning and contamination | 50 (43.5) | 65 (56.5) |
| Insects such as cockroaches and flies might transmit foodborne pathogens | 109 (94.8) | 6 (5.2) |
| Germs that make people sick grow well in cold temperatures | 17 (14.8) | 98 (85.2) |
| Learners can easily get sick from eating unsafe food | 111 (96.5) | 4 (3.5) |
| Foodborne outbreaks are natural life event | 37 (52.2) | 78 (67.8) |
| **Knowledge on personal hygiene** | | |
| Is it always necessary to wash hands after visiting a toilet? | 114 (99.1) | 1 (0.9) |

**Table 2.** *Cont.*

| Response Questions | Yes (%) | No (%) |
|---|---|---|
| Should you wash your hands even if you are using a spoon to handle food? | 99 (86.1) | 16 (13.9) |
| Would you wear jewellery while preparing food? | 16 (13.9) | 99 (86.1) |
| Would you dry hands by using kitchen towel? | 72 (68.6) | 43 (37.4) |
| Would you dry hands by using your apron after washing your hands? | 14 (12.2) | 101 (87.8) |

The findings of this study demonstrated VFHs' high level of knowledge on food safety and handling practices. For example, when VFHs were asked about the importance of having formal training on food safety and having a certificate on food safety, most of them affirmed that food safety knowledge was an important part of their job. Most of them also understood that unhygienic practices of food handling could cause sickness among the learners. These findings are corroborated by other study reports in South Africa [11,12] and the United Arab Emirates [16], where food handlers in schools and public service institutions demonstrated adequate hygiene knowledge in food handling. Thus, these study findings support the fact that formal training in food handling duties is a significant contributor to ensuring food safety practices. In contrast, in one South African study, food handlers in the NSNP lacked knowledge and awareness on many important aspects of microbial food hazards [2].

*3.3. Volunteer Food Handlers' Food Safety Practices*

In order to assess their safe food handling practices, food handlers were asked a number of questions, including whether they washed their hands before handling food, among others (Table 3). Reportedly, 87% washed their hands before handling food while the rest did not. Up to 70.4% of the food handlers washed their hands with soap and water, while the rest never did. Most of the food handlers (79%) covered their hair with hairnets, while the rest did not. Hairnets help to secure hair on peoples' heads and prevent the spread of micro-organisms into food.

**Table 3.** Volunteer food handlers' food safety practices (n = 115).

| Practice Question | No (%) | Yes (%) |
|---|---|---|
| **Manner of washing hands** | | |
| Often washed hands before handling food | 15 (13.1) | 100 (87) |
| Often rinsed hands with water only | 50 (43.5) | 65 (56.5) |
| Often used soap and water when washing your hands | 34 (29.6) | 81 (70.4) |
| Always washed hands with soap and water before handling food | 15 (13.0) | 100 (87.0) |
| **Manner of drying hands** | | |
| Often used a wet cloth to wipe hands | 61 (53.0) | 54 (47.0) |
| Often used paper or hand towel to dry hands | 82 (80.0) | 23 (20.0) |
| Often used kitchen or drying cloth to dry hands | 150 (75.7) | 28 (24.3) |
| **Washing learners' plates** | | |
| Washed learners' plates with soap and water | 60 (52.2) | 55 (47.8) |
| Washed learners' plates with water only | 95 (91.4) | 10 (8.6) |
| **Food preparation practices** | | |
| Often sorted, checked and separated foreign objects | 14 (11.4) | 101 (88.6) |
| Often washed fruits before serving | 22 (19.1) | 93 (80.9) |
| Often washed vegetables before cutting | 40 (34.8) | 75 (65.2) |
| Used same cloth to wash and dust | 58 (50.4) | 57 (49.6) |
| Often cooked food a day before | 78 (67.8) | 37 (32.2) |
| Often served food immediately after cooking | 61 (53.0) | 54 (47.0) |
| Smoked when preparing food | 111 (96.5) | 4 (3.5) |
| Often wore personal protective clothing | 105 (91.3) | 10 (8.7) |
| Often wore jewellery during food preparation | 101 (87.8) | 14 (12.2) |

When asked about the practice of washing learners' plates, the majority of respondents did (92.2%). When washing the plates, VFHs would use soap and water (47.8%) or water only (8.6%). Most of the food handlers cooked food on the same feeding day (67.8%), while the rest did not. Cooking food on the same day of serving is a correct practice, as it reduces the risk of microbial contamination. Most food handlers (89.0%) were compliant on the practice of sorting, checking and separating foreign objects in raw food before cooking. In addition, most food handlers never smoked during food preparation (93.9%, n = 108), while most of them also washed fruit and vegetables before use (87.0%). In the category of those who wore protective clothing (aprons), VFHs with 12 or less months of work experience were 66% were more likely to wear aprons than those with more than 12 months' work experience as VFHs (OR: 0.34, 95%CI: 0.12–0.91, *p* = 0.033). The newly recruited VFHs usually obtain basic training before embarking into their voluntary work and could easily remember to wear aprons more frequently than their colleagues who were trained less recently. In addition, the VFHs who joined the voluntary work much more recently were likely to have higher level of education and better awareness of food safety [17]. Women were 90% more likely to wear aprons than men (OR = 0.11; 95 CI: 0.03–0.45; *p* = 0.002). Plausibly, women work in the kitchen more than men do, particularly at home, and therefore it would be easy for them to wear aprons [12].

The results also showed that VHFs, who were aware that pathogens can be found in the skin of a healthy-looking food handler, were 2.5 times more likely to wear protective clothing (OR = 2.5, 95 CI: 1.01–6.09, *p* = 0.047). They can also be potential carriers of illness through unhygienic practices, which can contribute to learners' illnesses [17,18]. The current study reveals that food handlers were mindful of food safety practices, such as washing hands and covering hair. It is known that food handlers working in school feeding programmes can be the potential contributors to food poisoning risks and incidences, arising from their food handling practices [12].

This study's findings demonstrated a satisfactory level of safe food handling practices by VFHs. This was deduced by the proportion of participants who correctly responded to the statements or questions posed to them in the questionnaire tool. For example, over 80% of them washed their hands before handling food, while over 60% washed hands with soap and water, covered their hair with hairnets while preparing food and cooked food on the same feeding day. This finding corroborates our earlier study, where food vendors had their hair covered the whole time they prepared and served food [18]. However, there were instances where their safe food handling practices lapsed. For example, over 50% of participants rinsed hands with water only as they were working. It could be speculated that these schools are resource scarce and located in economically disadvantaged communities and could therefore not always afford to buy soap. Scarcity of resources has been identified as a big contributor to non-compliance to food safety practices [13,19]. Therefore, a gap exists where food safety knowledge is not translating to delivery of safe meals, and this could be the reason why food poisoning cases are still rife in the NSNP schools. Additionally, a number of factors could be attributed to the lapse in VFHs' safe practices, and may include forgetfulness, being too busy, lack of knowledge and culture of the workplace, which is often complacent with negative safety practices [20,21].

### 3.4. Observation of Facilities and Resources

In order to assess the condition of the facilities and availability of resources, a walk-through of the facilities used to prepare and serve food to the learners was conducted (Table 4). The checklist showed that most of the schools (97.4%) had some form of storage enclosures for NSNP food. Although most of the NSNP storage facilities were dedicated to the school feeding programme, a number of other school venues, including bookstores, classrooms, and to some extent, the principal's offices, were utilised for NSNP food storage. These facilities were not designed for food storage. For example, while some storage facilities had windows (79.1%, n = 91), the rest did not. Similarly, they had kitchen facilities for food preparation, but the inbuilt kitchens were poorly designed, and some schools

served by the NSNP programme subscribed to using mobile kitchens. It is possible that in the course of the programme roll-out, resources to implement the programme were not forthcoming. Thus, the schools had to resort to using any kind of infrastructure and facility that was available to be able to have the programme running. These shortcomings have negative implications for food safety, as improperly stored food can easily become spoilt and contaminated by disease-causing bacteria and moulds, and thus contribute to food-borne illnesses [18]. Reportedly, South African infrastructure design and kitchen facilities in the schools with feeding schemes are not designed for meal preparation [12]. This is an area of concern that needs immediate attention by the department of basic education.

**Table 4.** Resources available in the food preparation facilities at schools.

| Facility | Yes | No |
|---|---|---|
| Dining hall for learners | 2(1.7) | 113(98.3) |
| Food storage enclosures available | 112(97.4) | 3(2.6) |
| Storage has windows | 91(79.1) | 24(20.9) |
| Storage has shelves | 49(42.6) | 66(57.4) |
| Freezers available | 85(73.9) | 30(26.1) |
| Cold room available | 4(3.5) | 111(96.5) |
| Refrigerator available | 20(17.4) | 95(82.6) |
| Hot running water available | 24(20.9) | 91(79.1) |
| Water taps and basins available | 62(53.9) | 53(46.1) |
| Changing room available | 6(5.2) | 109(94.8) |
| Showers available | 0 | 115(100) |

In this current study, only 1.7% of the schools had a dining hall where learners could sit and dine. Although freezers were available in most schools (73.9%), refrigerators and cold rooms were scarce (4–20%). This deficiency may lead to food spoilage and contamination. In addition, just over half of schools (53.9%) had water taps and basins, while almost all schools had no changing rooms and showers for food handlers. This anomaly has huge implications for VFHs' food hygiene, as it makes it difficult for them to maintain personal cleanliness. Table 4 shows the availability of resources and facilities for food handling and preparation in the participating schools. The findings in our study resonate well with those reported by Mafuku et al. [13] and Uduku [14], showing that in South Africa, infrastructure design and kitchen facilities in the schools with feeding scheme programs were not designed for meal preparation. Such unsuitable facilities can lead to food contamination [19].

The checklist also gathered information on the type of structure of cooking areas. Most of the facilities (69.6%) were in-built using brick and mortar. Mobile containers were the second most used facilities (20.9%), while in some instances, classrooms and shacks were converted into cooking areas. In some extreme circumstances, cooking was performed in open spaces (2.6%) and *shacks*, a local term for temporary shelters (6.1%). This has real safety implications for food safety, given that food cooked in open spaces can be exposed to a number of environmental hazards such as dust and air pollutants, including disease-causing microorganisms [4,14].

## 4. Study Limitations

One of the main limitations in this study is that the study population sample was from one district (Gauteng North District) amongst all the districts within Gauteng Province. This is the smallest district in terms of the number of schools. This may limit the scope of the study findings. Secondly, the study was conducted in one Gauteng region, and observations and findings could be different from other regions or settings in the country, or in different countries which run similar programs.

## 5. Conclusions

This study has shown that respondents had sufficient levels of knowledge on food safety and safe food handling practices. However, there was a lapse in some of the safe practices and this could be attributed to annual turnover of VFHs and inadequate facilities and resources. Thus, there is a need to further support VFHs to comply with recommended safe food handling practices.

**Author Contributions:** P.K.C., M.R. both contributed to the conceptualisation of the study. M.R. was involved data collection, analysis and reporting. P.K.C. supervised the study and performed data analysis and write-up of the manuscript. All authors have read and agreed to the published version of the manuscript.

**Funding:** This research received no external funding.

**Institutional Review Board Statement:** The study was conducted in accordance with the Declaration of Helsinki, and approved by the Institutional Review Board—Sefako Makgatho University Research Ethics Committee (SMUREC/H/287/2016:PG).

**Informed Consent Statement:** Informed consent was obtained from all subjects involved in the study.

**Data Availability Statement:** Data supporting reported results can be obtained from the corresponding author upon request.

**Acknowledgments:** We acknowledge the willingness of volunteer food handlers in the Tshwane community to participate in this study.

**Conflicts of Interest:** The authors declare no conflict of interest.

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
