# Peer review of "Volunteer Food Handlers’ Safety Knowledge and Practices in Implementing National School Nutrition Programme in Gauteng North District, South Africa"

_safety, 2022_

Round 1

Reviewer 1 Report

The publication is very interesting and addresses the extremely important issue of the safety of food served in schools. Generally, it is of a good standard.

However, the authors did not provide the results of the statistical evaluation, although the methods state that it will be performed.

Please report the results of the statistical analysis and use them to evaluate and discuss the results. They should also be used in conclusions.

Author Response

The publication is very interesting and addresses the extremely important issue of the safety of food served in schools. Generally, it is of a good standard.

However, the authors did not provide the results of the statistical evaluation, although the methods state that it will be performed. Statistical evaluation included.

Please report the results of the statistical analysis and use them to evaluate and discuss the results. They should also be used in conclusions. Statistical analysis results included in the discussion.

Reviewer 2 Report

  1. The paper titled “Volunteer food handlers’ safety knowledge and practices in schools implementing national school nutrition programme in Gauteng North district, South Africa” described study investigated food safety knowledge and practices of VFHs working for the National School Nutrition Programme (NSNP), Gauteng North District, South Africa.. The manuscript has potential however need major revision for improvement
  2. Title should be like “Volunteer food handlers’ safety knowledge and practices in implementing national school nutrition programme in Gauteng North district, South Africa”
  3. Abstract is fine however, improve language and consistency of study.
  4. Introduction is written poor as lack background of study so add paragraphs regarding food safety concerns and program significance for the study
  5. Material and method section need improvement as how the Performa was developed and was it validated?
  6. Elaborate the inclusion and exclusion criteria for study participants
  7. Result written in good way however add latest references that can support the findings of the study
  8. In conclusion major focus on findings with practical application of study?
  9. Results reported food safety incidents but conclusions documented that handlers has food safety knowledge and practices that seems inconsistent. Please justify  
  10. Grammatical mistakes observed so need to go through the paper for language and grammatical mistakes

Author Response

  1. Title should be like “Volunteer food handlers’ safety knowledge and practices in implementing national school nutrition programme in Gauteng North district, South Africa” Title amended to improve it.
  2. Abstract is fine however, improve language and consistency of study. Abstract amended for language and consistency.
  3. Introduction is written poor as lack background of study so add paragraphs regarding food safety concerns and program significance for the study. Introduction improved.
  4. Material and method section need improvement as how the Performa was developed and was it validated? I don’t understand your suggestion but if it is about the tool, it has been improved.
  5. Elaborate the inclusion and exclusion criteria for study participants. Inclusion criteria addressed.
  6. Result written in good way however add latest references that can support the findings of the study. Additional references added where necessary/possible.
  7. In conclusion major focus on findings with practical application of study? Practical aspects of study included in conclusion
  8. Results reported food safety incidents but conclusions documented that handlers has food safety knowledge and practices that seems inconsistent. Please justify . Inconsistency addressed. 
  9. Grammatical mistakes observed so need to go through the paper for language and grammatical mistakes. Grammatical mistakes attended to.

Reviewer 3 Report

Dear authors,

The study is necessary to map the conditions of school food production in South Africa since the program is necessary for learners. However major improvements are necessary for the manuscript.

Abstract - conclusion must be improved. The results do not conduct to these conclusions.

Introduction - Do only volunteers work in preparing and serving food at school? Do schools have permanent workers that can lead volunteers?

Line 45 - there have been reports of food poisoning outbreaks in public schools

Materials and methods

line 73 - Food handlers working in the selected schools - from the 38 schools, dis you select some? This is not clear. or did you visit all the 38 schools?

Were the instruments validated? or did you select validated instrument and puth them together for this research?

Do the instrument have a score for classification?

Results

This is results and discussion section, not only results.

How many volunteers did not accept to participate? it is important to have the participation rate

Work experience - is it in food production?

lines 107 and 108 - repeated information

workshop - how many hours of training?

lines 113 to 120 - it is not a discussion for the sociodemographic, change order

Table 1 - the% relating to marital status is not correct

line 189 to 191 - repeated information

lines 206, 207 brings that food handlers have adequate hygiene practices

One thing is to evaluate knowledge, other is to evaluate practices

Table 2 - It is necessary to put on gloves when cooking. - What type of gloves? in all the cooking processes? What is the correct answer for this question?

Question - If s food handler has an infected cut ... this does not lead to automatically vomiting and diarhrea. This question is confusing.

Much more discussion is needed with results from table 2.

line 221 - is it table 2?

line 235 - rewrite

line 242 - how did you classify practices as satisfactory? Was there a score for the instrument?

Much more discussion of Table 3

Many questions and responses could be crossed for more accurate discussion.

Where did the checklist come from? Is it validated? Does it have a score?

line 273 - had a dining hall

line 274 - rewrite

275 - take of "in"

Figure 1 is not necessary - insert % in the text

Do food handlers have their own bathroom? Was it close to the cooking area?

Discussion on the checklist is needed.

4 - limitations - validation of instruments

Conclusion are not supported by the results

Author Response

The study is necessary to map the conditions of school food production in South Africa since the program is necessary for learners. However major improvements are necessary for the manuscript.

Abstract - conclusion must be improved. The results do not conduct to these conclusions. Abstract conclusion improved

Introduction - Do only volunteers work in preparing and serving food at school? Do schools have permanent workers that can lead volunteers? Information included in the introduction

Line 45 - there have been reports of food poisoning outbreaks in public schools corrected

Materials and methods

line 73 - Food handlers working in the selected schools - from the 38 schools, dis you select some? This is not clear. or did you visit all the 38 schools? All Schools were visited. Sentence amended.

Were the instruments validated? or did you select validated instrument and puth them together for this research? The tool was designed based on studies which used validated tools and have been cited.

Do the instrument have a score for classification? Percentage proportions and frequencies were used to judge the level of knowledge and practices.

Results

This is results and discussion section, not only results. Subheading corrected.

How many volunteers did not accept to participate? it is important to have the participation rate. All participants were approached participated in the study. However 115 out of 144 participated. Response rate included in results.

Work experience - is it in food production? Work experience in the voluntary work of serving food to school learners

lines 107 and 108 - repeated information Repeated information deleted

workshop - how many hours of training? Duration of training explained

lines 113 to 120 - it is not a discussion for the sociodemographic, change order Order changed

Table 1 - the% relating to marital status is not correct % corrected

line 189 to 191 - repeated information Repeated information deleted

lines 206, 207 brings that food handlers have adequate hygiene practices Sentence corrected

One thing is to evaluate knowledge, other is to evaluate practices

Table 2 - It is necessary to put on gloves when cooking. - What type of gloves? in all the cooking processes? What is the correct answer for this question? Kitchen gloves. Correct answer: It is not necessary. Kitchen gloves are mainly worn when handling hot objects.

Question - If s food handler has an infected cut ... this does not lead to automatically vomiting and diarhrea. This question is confusing. Question excluded from results

Much more discussion is needed with results from table 2. More discussion given

line 221 - is it table 2? Table 3

line 235 – rewrite. Re-written

line 242 - how did you classify practices as satisfactory? Was there a score for the instrument? This was judged by the proportion of participants that correctly responded to the statements or questions posed to them in the  questionnaire tool. Over 50% correct responses was considered satisfactory.

Much more discussion of Table 3. Done

Many questions and responses could be crossed for more accurate discussion.

Where did the checklist come from? Is it validated? Does it have a score? The tools (questionnaire and checklist) were designed based on validated methodologies in the literature that undertook similar studies and they are cited. No scoring of checklist.

line 273 - had a dining hall corrected

line 274 – rewrite. Sentence re-written

275 - take of "in" Removed

Figure 1 is not necessary - insert % in the text Figure removed

Do food handlers have their own bathroom? Was it close to the cooking area? No changing rooms or showers were available.

Discussion on the checklist is needed. Discussion improved

4 - limitations - validation of instruments Addressed

Conclusion are not supported by the results. Conclusion amended.

Reviewer 4 Report

The authors studied the food safety knowledge and practices of the VFHs in schools implementing a national school nutrition programme in cauteng North district, South Africa and observed the facilities' conditions and available resources.

The originality of this study is limited to the country where it has been conducted as It did not add significant knowledge to the literature with regards to the contributing factors to food handlers' knowledge and practices in food safety.

The manuscript reads well and is generally well written except that it needs a thorough revision for minor editing, correcting punctuations and eliminating spaces. However, the authors have not described in their methods how they validated the questionnaire and whether it is a reliable tool. The survey tool should be validated and tested for its reliability unless the authors adapted a validated tool, but that was not the case. The authors referred to literature and that is not enough. The questionnaire design, particularly the section on food safety practices, is not fit in terms of its content to generate reliable information unless tested. For instance, the corresponding information in Table 3 is confusing. It shows “ Always washed hands with soap and water before handling food” and “Often used soap and water when washing your hands”, I don’t see a difference in both. If they always wash their hands with soaps before preparing food, then how is it that fewer of them “often” washed their hands with soap and water?

what is the difference between the “Often washed learners’ plates” and the other two related sentences?

Minor corrections for editing are shown below but not a comprehensive list:

Eliminate space in L 34, L179, 193,L201, L243, 253, 254 (this is not an exhaustive listing)

L 109: attendance in instead of “of”

L124: volunteering

L124: remove the comma after counterparts

L125: this makes

L127: in the vicinity of

L220 and L222: add “their” hands

L222: a comma after water

L178: a comma after practices

L235: sentence structure.

Table 2: one of the sentences has a question mark but not formulated as a question. Another is formulated as a question but has no question mark. The statement structure should be reviewed and corrected.

Table 4: ensure consistency in the way the conditions are described. For instance, instead of “has freezer” you could use “freezer availability” , instead of "having a dining hall", use "Dining hall for staff".

L251-252: It is unlikely to be cultural. How usual is it to wear PPE at home when preparing food in the US or European countries where we may see compliance to wearing the PPE? The authors may need to discuss this observation from the regulatory point of view, knowledge and awareness, attitudes. I recommend removing the "cultural" aspect in your explanation of this observation.

Please remove the word “disappointingly” from the paper (L273 and elsewhere)

L292: contaminants instead of “insults”.

Author Response

The manuscript reads well and is generally well written except that it needs a thorough revision for minor editing, correcting punctuations and eliminating spaces. However, the authors have not described in their methods how they validated the questionnaire and whether it is a reliable tool. The survey tool should be validated and tested for its reliability unless the authors adapted a validated tool, but that was not the case. The authors referred to literature and that is not enough. The questionnaire design, particularly the section on food safety practices, is not fit in terms of its content to generate reliable information unless tested. For instance, the corresponding information in Table 3 is confusing. It shows “ Always washed hands with soap and water before handling food” and “Often used soap and water when washing your hands”, I don’t see a difference in both. If they always wash their hands with soaps before preparing food, then how is it that fewer of them “often” washed their hands with soap and water? These are different manners of washing hands. The risk of food contamination can vary by each manner of washing.

what is the difference between the “Often washed learners’ plates” and the other two related sentences? The section has been amended to be more informative. The health risks are multiplied if soap was not used when washing plates.

Minor corrections for editing are shown below but not a comprehensive list:Eliminate space in L 34, L179, 193,L201, L243, 253, 254 (this is not an exhaustive listing) Corrected

L 109: attendance in instead of “of” corrected

L124: volunteering - Corrected

L124: remove the comma after counterparts Done

L125: this makes Corrected

L127: in the vicinity of  Corrected

L220 and L222: add “their” hands Corrected

L222: a comma after water. Corrected

L178: a comma after practices. corrected

L235: sentence structure. Corrected

Table 2: one of the sentences has a question mark but not formulated as a question. Another is formulated as a question but has no question mark. The statement structure should be reviewed and corrected. Corrected
Table 4: ensure consistency in the way the conditions are described. For instance, instead of “has freezer” you could use “freezer availability” , instead of "having a dining hall", use "Dining hall for staff".Corrected
L251-252: It is unlikely to be cultural. How usual is it to wear PPE at home when preparing food in the US or European countries where we may see compliance to wearing the PPE? The authors may need to discuss this observation from the regulatory point of view, knowledge and awareness, attitudes. I recommend removing the "cultural" aspect in your explanation of this observation. Removed
Please remove the word “disappointingly” from the paper (L273 and elsewhere). Removed

L292: contaminants instead of “insults”. Replaced “insults”.

Round 2

Reviewer 1 Report

No other comments

Reviewer 2 Report

The suggested changes are incorporated so paper may be considered for publication

Reviewer 3 Report

The authors addressed all the recommendations.